# Inhibitory Effect of Hesperidin on the Expression of Programmed Death Ligand (PD-L1) in Breast Cancer

**DOI:** 10.3390/molecules25020252

**Published:** 2020-01-08

**Authors:** Prachya Kongtawelert, Benjawan Wudtiwai, Thuzar Hla Shwe, Peraphan Pothacharoen, Thanyaluck Phitak

**Affiliations:** Thailand Excellence Center for Tissue Engineering and Stem Cells, Department of Biochemistry, Faculty of Medicine, Chiang Mai University, Chiang Mai 50200, Thailand; Benjawanwudtiwai@gmail.com (B.W.); thuzar.hs@gmail.com (T.H.S.); peraphan.pothacharoen@gmail.com (P.P.); thanyaluck.phitak@cmu.ac.th (T.P.)

**Keywords:** breast cancer, programmed death ligand 1, Akt, immune checkpoint, hesperidin

## Abstract

Programmed death ligand 1 (PD-L1) is overexpressed in the most aggressive breast cancer subtype, triple-negative breast cancer (TNBC), assisting the eradication of antitumor immunity, and thereby enhancing the survival of the tumor. This study explored how hesperidin affects PD-L1 expression, and thereby cancer progression in breast cancer cells. We found that MDA-MB231, the triple-negative breast adenocarcinoma cancer cell line, (high aggressiveness) has higher expression, in both mRNA and protein, of PD-L1 than that of the other breast cancer cell line, MCF-7 (low aggressiveness). Hesperidin inhibited cell proliferation in MDA-MB231 cells. Additionally, high expression of PD-L1 (both mRNA and protein) in aggressive cancer cells was strongly inhibited by hesperidin through inhibition of Akt and NF-κB signaling. Moreover, hesperidin treatment, by inhibiting activation of matrix metalloproteinases such as MMP-9 and MMP-2, suppressed the metastatic phenotype and cell migration in the PD-L1 high-expressing MDA-MB231 cells. In summary, hesperidin inhibits breast cancer cell growth through the inhibition of the expression of PD-L1 via downregulation of Akt and NF-κB signaling in TNBC. Moreover, hesperidin significantly suppresses cell migration of MDA-MB231 cells. Our findings reveal fresh insights into the anticancer effects of hesperidin which might have potential clinical implications.

## 1. Introduction

One of the common life-threatening cancers in women is breast cancer. Triple-negative breast cancer (TNBC) is a breast cancer subtype which is heterogeneous in nature and has the characteristic of negativity for estrogen receptor (ER), progesterone receptor (PR), and epidermal growth factor receptor (EGFR or HER2). It has aggressive metastatic potential and, as a consequence, results in a poor prognosis. It is resistant to many therapies for breast cancer including hormone therapy, chemotherapy, and surgical excision [1]. Thus, a new treatment is required and there is a need for further in-depth investigations of this emerging subtype of breast cancer.

Programmed death ligand 1 (PD-L1) is a preferred ligand for programmed death receptor 1 (PD-1) and is expressed on cancer cells. Its expression in cancer cells is induced by many factors in the inflammatory tumor microenvironment including EMT, cytokines, mainly IFN-γ, and growth factors such as EGF [2]. Activated T cells express the PD-1 receptor [3] and the interaction between PD-L1 and PD-1 elaborates immune evasion modulation via inhibition of T cell function through inducing apoptosis of the T cells [4]. The expression of PD-L1 in several cancer types leads to immune escape, drug-resistance, and metastasis via the signaling pathways such as ERK and PI3K/Akt signaling. PD-L1/PD-1 is a target for cancer treatment and many reports have shown that drugs targeting PD-L1 and PD-1 are effective in many types of cancer [5]. For this reason, the discovery of natural compounds that can block the PD-L1/PD-1 signaling in cancer cells is an interesting and promising avenue for research.

Hesperidin is a flavonoid amply found in citrus species including orange peel which is broadly used in Chinese herbal medicine. Previous reports have shown that hesperidin is nontoxic to normal cells, but it is an effective anticancer agent against several types of cancer including bladder cancer, prostate cancer, breast cancer, and hepatocellular carcinoma [6,7,8]. It has already been reported that hesperidin induces anticancer activity through the promotion of apoptosis [9]. A recent report showed that hesperidin also inhibits epithelial–mesenchymal transition (EMT) and the expression of members of the MMP family, and thereby inhibiting cell invasion and contributing to its anticancer activity [10]. However, the role of hesperidin in inhibiting PD-L1 mediated immune checkpoint and ultimately increasing the tumor immune response, has yet to be investigated in breast cancer.

This study investigated the changes in the expression of PD-L1 elicited by hesperidin, as well as the underlying mechanisms in the MDA-MB231 cell line. We found that the expression of PD-L1 is decreased by hesperidin via inhibiting the PI3K/Akt and NF-κB pathway in the MDA-MB231 cell line. This study also investigated whether hesperidin prevents metastasis phenotypes by reducing PD-L1 expression of the MDA-MB231 cells and found that hesperidin acts as an inhibitor of the immune checkpoint, PD-L1, in breast cancer development.

## 2. Results

### 2.1. The PD-L1 Expression Level Is Associated with Aggressive Breast Cancer Cell Lines

It has been reported that PD-L1 is overexpressed in TNBC cells such as MDA-MB231 [11]. In this study, the presence of PD-L1 on the MDA-MB231 cell surface was confirmed (Figure 1A,B). The flow cytometry result showed high expression of PD-L1 on the MDA-MB231 cell surface as compared with a nonaggressive breast cancer cell line (MCF-7). The qRT-PCR analysis demonstrated that MDA-MB231 expressed higher PD-L1 mRNA and PD-L1 protein as compared with MCF-7 (Figure 1C,D). Taken together, the data confirm that PD-L1 is overexpressed in the MDA-MB231 subtype of TNBC.

### 2.2. Hesperidin Inhibits MDA-MB231 Cells Viability

The chemical structure of hesperidin is shown in Figure 2A. The anticancer effects of hesperidin have been reported previously [6,12]. To confirm the cytotoxic effect of hesperidin on MDA-MB231, MTT assay was performed at 24, 48, and 72 h after hesperidin treatment. The results showed that hesperidin significantly decreased cell viability as compared with the control group. The 20% inhibitory concentrations (IC_20_) of hesperidin in MDA-MB231 after 24, 48, and 72 h were approximately 118.18, 94.00, and 72.67 µM, respectively, demonstrating that the ability of hesperidin to inhibit cell proliferation is dose and time dependent (Figure 2B). The nontoxic concentrations of hesperidin (0, 10, 20, 30, 40, and 50 µM) at 48 h were applied in the next experiments.

### 2.3. Hesperidin Decreases PD-L1 Expression in MDA-MB231 Cells

It is a well-known fact that PD-L1 expression in cancer cells helps protect the cells from immune-mediated surveillance [13]. In this study, the effects of hesperidin on high-expressing PD-L1 MDA-MB231 cells were first determined. The levels of mRNA and protein expression of PD-L1 were dose-dependently inhibited by hesperidin, i.e., decreased by 50% at 24.17 µM and 33.18 µM concentrations, respectively (Figure 3A,B). These findings suggest that hesperidin dose-dependently inhibits both PD-L1 mRNA and protein.

### 2.4. Hesperidin Decreases PD-L1 by Downregulating Akt and NF-κB in MDA-MB231 Cells

A previous study described several mechanisms controlling PD-L1 expression in breast cancer cells [14]. One important mechanism is EMT progression, which is demonstrated to upregulate PD-L1 expression in breast cancer cells. The PI3K/Akt, ERK/MAPK, SMAD, and NF-κB signaling pathways are those reported to account for the EMT process [15]. In cancer, PI3K/AKT is essential for the EMT-associated enhanced migration [16], whereas NF-κB is implicated in the chemoresistance induced by EMT [17]. We observed that both the PI3K inhibitor, LY294002, and the NF-κB inhibitor, BAY11-7082, inhibited PD-L1 expression in PD-L1 high expressing MDA-MB231 cells (Figure 4C,D). These results imply that these two pathways, the Akt and NF-κB pathways, are involved in PD-L1 expression in high expressing MDA-MB231 cells. Moreover, hesperidin treatment (10 to 50 µM) as compared with the control group, resulted in significant inhibition of expression of PD-L1, and the proteins of signaling pathways, p-Akt, p-p65, and p-ERK (Figure 4A,B and Appendix A). These findings suggest that PD-L1 is an upregulator of breast cancer progression while hesperidin delays this process by suppressing the Akt and NF-κB signaling pathways.

### 2.5. Hesperidin Suppresses Migration and MMP Secretion in High-Expressing PD-L1 MDA-MB231 Cells

The immune escape ability of cancer cells is a key factor in the successful metastasis of cancer cells. An important immune escape mechanism of cancer cells involves PD-L1 binding to PD-1 [18]. In addition, the activated MAPK/ERK and PI3K/AKT pathways are reported to be responsible for the PD-L1/PD-1-induced metastasis [19]. In this study, we studied whether hesperidin, at non-toxic concentrations, diminished PD-L1 induced metastatic phenotypes of MDA-MB231 cells, through inhibition of cell migration and secretion of matrix metalloproteinases, MMP-9 and MMP-2. We found that hesperidin significantly reduced the levels of MMP-9 and MMP-2 secreted from PD-L1 high-expressing MDA-MB231 cells (Figure 5A,B). In a wound healing assay in this study, the migration abilities of PD-L1 high-expressing MDA-MB231 cells were shown to be decreased by hesperidin (Figure 5C). The relationship between PD-L1 expression, MMP secretion, and metastatic phenotypes can also explain how hesperidin can abrogate the metastasis phenotypes through suppressing MMP-9 and MMP-2 secretion and inhibiting migration in PD-L1 high-expressing MDA-MB231 cells.

## 3. Discussion

PD-L1 is expressed mainly on the surface of antigen presenting cells such as dendritic cells, macrophages, and many types of cancer cells [4], for example, breast cancer [14], hepatocellular carcinoma [20], and ovarian cancer [21]. PD-L1 on the cancer cells protects them from being recognized by the host immune system [22] and developing immune evasion which in turn leads to cancer progression, invasion, and metastasis [23]. It was recently established that upregulated status of PD-L1 is related to metastasis [24]. Breast cancer which lacks hormone receptors, estrogen and progesterone receptors, and HER2 protein (hormone receptor negativity) is known as triple-negative breast cancer (TNBC). A study found that this aggressive subtype of basal breast cancer exhibits overexpression of PD-L1 and notably metastatic phenotype. That study also found that EMT, one of the pro-metastatic processes, induces the expression of PD-L1 in TNBC cells [14]. This study used flow cytometry, qRT-PCR, and Western blotting to confirm that PD-L1 is overexpressed in TNBC (MDA-MB231) but not in MCF-7 (a luminal subtype group), a result consistent with previously reported findings.

Hesperidin has been shown to have anticancer effects promoting apoptosis in liver cancer [6] and bladder cancer [25]. Moreover, hesperidin inhibits the expression of MMP [10] and epithelial–mesenchymal transition (EMT) related proteins, suppressing cell migration and invasion [26], as well as being an anti-inflammatory [2]. In this study, we discovered that hesperidin decreases the expression of PD-L1 in cancer cells which contributes to tumor immune evasion. This was the first demonstration that hesperidin at non-toxic concentrations (10 to 50 µM) strongly suppresses PD-L1 expression in excessive PD-L1-expressing cells (MDA-MB231) as well as both mRNA and protein levels, indicating that hesperidin acts as an immune checkpoint inhibitor by targeting PD-L1 in MDA-MB231 cells.

Previous studies have described that several mechanisms, including loss of phosphatase and tensin homologue (PTEN) and overexpression of PI3K activation, are involved in regulating PD-L1 expression in breast cancer [27]; and EMT progression induces PD-L1 expression associated metastatic phenotypes. The major signaling pathways of EMT progression are PI3K/AKT, ERK, SMAD, and NF-κB [15]. We confirmed the expression of PD-L1 is via the PI3K/AKT and NF-κB pathways, a finding consistent with the previous study. Interestingly, hesperidin inhibited the NF-κB p65 and p-Akt expression significantly. Both of NF-κB and p-Akt are known to regulate the expression of PD-L1 and are also related to metastasis.

Finally, we investigated the relationship between PD-L1 and the metastatic phenotype. A previous study stated that both mRNA and protein levels of MMP2 in tumor tissues were decreased through the PD-L1 blockade [28]. Our study also exhibited that hesperidin can suppress PD-L1 expression, decrease MMP-9 and MMP-2 secretion, and inhibit cell migration in a dose-dependent manner (EC50 in Appendix A), signifying that hesperidin reduces the expression of PD-L1 and suppresses metastatic phenotypes in PD-L1 high-expressing cells through the inhibition of the NF-κB and Akt pathways (Figure 6). Our results suggest that a phytochemical such as hesperidin can also act as an antitumor agent by targeting the immune checkpoint PD-L1.

The pharmacokinetic properties of hesperidin after ingestion include the transformation of hesperidin to hesperetin by β-glucosidases of the microorganisms in the ileum, the colon and the liver by cleaving flavonoid glycosides before absorption and it significantly increases the bioavailability of hesperidin [29]. Although bioavailability of orally administered hesperidin is low, structural modification or nanotechnology may be able to address this limitation. Future in vivo studies should be conducted.

There certainly are some limitations in the present study. First, the relationship between PD-L1 and metastasis was not investigated, although it is reported that PD-L1 is correlated with metastasis in breast cancer cells [14]. The second limitation is that the reproducibility of these data in other PD-L1 expressing breast cancer cell lines was not investigated. Because breast cancer cell lines have a variety of endogenous biological differences, more than one cell line should be involved in future studies. The third limitation is that the level of PD-L1 protein was not evaluated in the metastatic setting wherein the trials of PD-L1 inhibitors were conducted. However, our results do suggest that MDA-MB231 with high PD-L1 expression exhibits excessive metastasis and that PD-L1-targeted immunotherapy might improve the efficacy of treatment.

## 4. Materials and Methods

### 4.1. Reagents

Dulbecco’s Modified Eagle’s medium-high glucose (DMEM-HG), fetal bovine serum (FBS), trypsin-EDTA solution, and phosphate-buffered saline (PBS) were purchased from Gibco (Grand Island, NY, USA). Hesperidin, 3-(4,5-dimethythiazol-2-yl)-2,5-diphenyltetrazolium bromide (MTT), and dimethyl sulfoxide (DMSO) were purchased from Sigma Chemical, Inc. (St Louis, MO, USA). Primary antibodies against PD-L1, AKT, pAKT (ser473), pp65, p65, and actin, as well as horse-radish peroxidase (HRP) conjugated anti-rabbit immunoglobulin G (IgG) were purchased from Cell Signaling Technology, Inc. (Beverly, MA, USA). SuperSignal Wes Femto Maximum Sensitivity substrate kit Pico and Restore^TM^ Plus Western Blot Stripping buffer were obtained from Thermo Fisher Scientific Inc. (Waltham, MA, USA). Protease inhibitor was obtained from Roche Diagnostics, (Mannheim, Germany). PI3K inhibitor (LY294002) and the IKK inhibitor (Bay 11-7082) were purchased from Calbiochem (Merk Darmstadt, Germany).

### 4.2. Cell Culture

Two human breast adenocarcinoma cell lines, MCF-7 and MDA-MB231, were cultured in Dulbecco’s Modified Eagle’s medium-high glucose (DMEM-HG) (Gibco, Grand Island, NY, USA), supplemented with serum, 10% fetal bovine serum (Gibco, Grand Island, NY, USA), and antibiotics penicillin/streptomycin (Gibco, Grand Island, NY, USA), in a humid atmosphere incubator with 5% CO_2_ at 37 °C. The cells were used for the relevant treatments or subcultured for further passages when their confluence reached 70% to 80%.

### 4.3. Cytotoxic Assay

The cytotoxicity of hesperidin on MDA-MB231 cells was determined using the MTT assay. The cells (5 × 10^3^ cells/well) were seeded in 96-well culture plates and incubated for either 24, 48, or 72 h after being treated with various concentrations of hesperidin. After incubation, MTT dye (Sigma-Aldrich, St. Louis, MO, USA), at a final concentration of 500 μg/mL, was added to the cells for 4 h at 37 °C in a CO_2_ incubator. After removing supernatant upon completion of 4 h incubation, the formed purple-formazan crystals were then dissolved in DMSO. Absorbance was measured using a spectrophotometric plate reader at 540 nm and a reference wavelength of 630 nm. Cell viability was calculated as, % cell survival = (OD of sample/OD of control) × 100. For each parameter, data were represented as mean ± SD. Experiments were carried out for three independent experiments and was triplicated in each time.

### 4.4. Real-Time PCR (RT PCR)

Using an illustra-RNAspin Mini RNA Isolation kit (GE Healthcare Europe GmbH, Freiburg, Germany), the total RNA from the cells was extracted and isolated. One microgram of tRNA was used to reverse transcribed to cDNA using an iScript™ cDNA Synthesis Kit (Bio-Rad, Hercules, CA, USA) according to the instructions from the manufacturers. Polymerase chain reactions were performed using an Applied Biosystems 7500/7500 Fast Real-Time PCR system and SensiFAST™ SYBR^®^Lo-ROX (Bio-Rad, Hercules, CA, USA). The primer sequences used in this study were as follows: human PD-L1 forward primer 5′-GGACAAGCAGTGACCATCAAG-3′;human PD-L1 reverse primer 5′-CCCAGAATTTACCAAAGTGAGTCCT-3′;human GAPDH forward primer 5′-TGGTATCGTGGAAGGACTCATGAC-3′;human GAPDH reverse primer 5′-ATGCCACTCAGCTTCCCGTTCAGC-3.′

The relative expression of each gene was normalized to an expression of GAPDH as an internal control.

### 4.5. Western Blotting Assay

A total of 2 × 10^5^ cells/well of MDA-MB231 were seeded in a culture plate (6-well plate) and treated with various concentrations of hesperidin (0 to 50 μM) for 48 h. After treatment, the cells were harvested and lysed with radioimmunoprecipitation assay (RIPA) buffer containing both protease and phosphatase inhibitors. The supernatants were collected, and the protein was subjected to 12% SDS-PAGE, and then, transferred to the nitrocellulose-type membranes from GE Healthcare Europe GmbH, Freiburg, Germany. Then, membranes were incubated with specific antibodies against interested signaling molecules, in both phosphorylated and total form, including PD-L1, Akt, and p65, followed by an appropriate secondary antibody. The loading level of β-actin was utilized as a control. The development of protein bands was done using SuperSignal West Femto Substrate (Thermo Scientific) and the bands were photographed using a molecular ChemiDoc XRS system (Bio-Rad); band density was analyzed with ImageJ software. After one target protein was detected, the primary and secondary antibodies complex was removed using the stripping buffer (Thermo Scientific). Then, the membrane was reprobed with the antibody against another interested protein and its level was determined again.

### 4.6. Flow Cytometry

Cells were stained with anti-PD-L1 antibody of 1:200 dilution in phosphate buffer saline (PBS) with 0.5% BSA and incubated for 30 min on ice. Cells were washed using PBS, and then, were incubated on ice with Alexa Fluor^®^ 647 goat anti-rabbit IgG for 30 min. The labeled single cells were analyzed for the expression status of cell-surface PD-L1 protein by using flow cytometry (FACSCAtia™III Cell Sorter; BD Biosciences, Franklin Lakes, NJ, USA), and median fluorescence intensity (MFI) was used to estimate the expression level.

### 4.7. Gelatin Zymography

Gelatin zymography was used to measure MMP-2 and -9 activities from MDA-MB231 cells. After the treatment, the conditioned medium was collected. The protein concentrations were determined using the Bradford method. Protein samples were separated in polyacrylamide gels (10%) supplemented with reagent grade gelatin (2 mg/mL). Gels were washed and incubated in activating buffer for 24 h. Next, the gels were stained with 0.5% Coomassie blue and destained (methanol:acetic acid:water). The area of protease appears as a clear band against the dark blue background where the protease has digested the substrate.

### 4.8. Statistical Analysis

Each experiment was independently repeated three times and mean ± SD values of the three repetitions were used. Statistical analysis to interpret the results was performed using SPSS software, including one-way ANOVA followed by the appropriate post hoc test such as Tukey HSD. *p* values < 0.05 were considered statistically significant.

## 5. Conclusions

Hesperidin actively suppresses NF-κB and Akt pathway activation, leading to decreased expression levels of PD-L1 in breast cancer cells. Restricting breast cancer growth and reducing the migratory potential in these cells, hesperidin attenuates their aggressiveness. These observations suggest novel insights into possible avenues for research relating to the anticancer properties of hesperidin with prospective clinical implications.

## Figures and Tables

**Figure 1 molecules-25-00252-f001:**
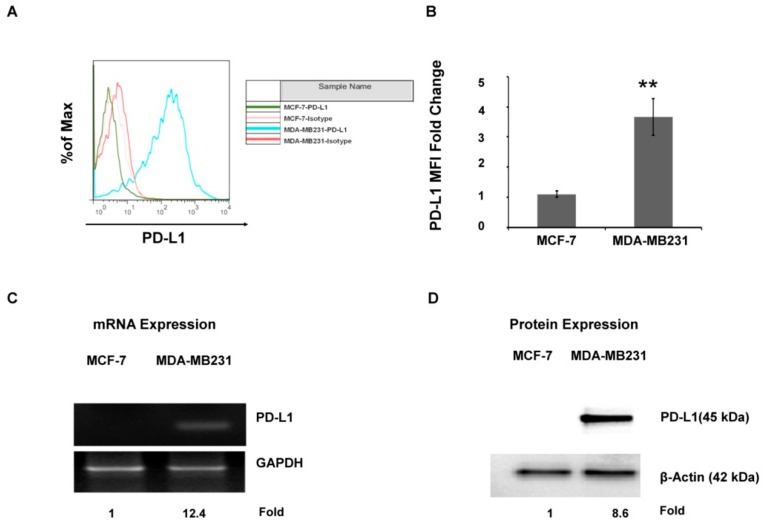
Programmed death ligand 1 (PD-L1) expression status of breast cancer cells: (**A**) Cell surface PD-L1 and (**B**) an average of the fold change of median fluorescence intensity (MFI). (**C**) PD-L1 mRNA expression and (**D**) PD-L1 protein expression. Each value is the mean ± SD of three independent experiments. Statistical significance ** *p* < 0.01.

**Figure 2 molecules-25-00252-f002:**
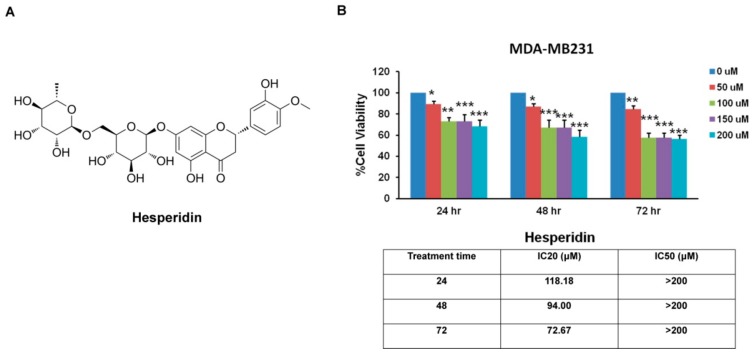
The cytotoxic effect of hesperidin assessed by MTT assay. (**A**) Chemical structure of hesperidin and (**B**) shows the percentage of cell viability of MDA-MB231 breast cancer cells, grown in the presence of hesperidin (0 to 200 μM) at 24, 48, and 72 h. All data are presented as mean ± SD from three or more independent experiments. Statistical significance * *p* < 0.05, ** *p* < 0.01, and *** *p* < 0.001 versus the control at equal incubation periods.

**Figure 3 molecules-25-00252-f003:**
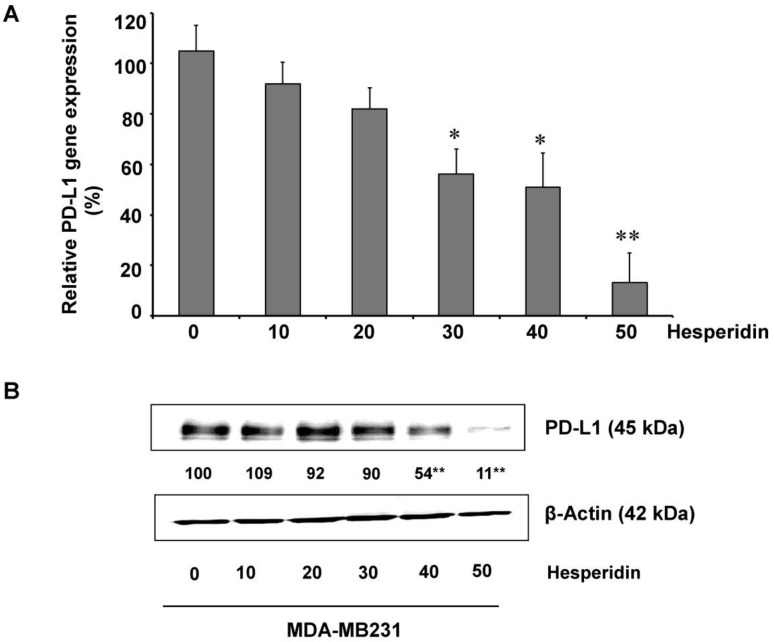
Inhibition of PD-L1 expression by hesperidin in MDA-MB231 cells: (**A**) PD-L1 mRNA expression and (**B**) protein levels of PD-L1 protein. Data indicated as mean ± SD of three independent experiments. Statistical significance * *p* < 0.05 and ** *p* < 0.01.

**Figure 4 molecules-25-00252-f004:**
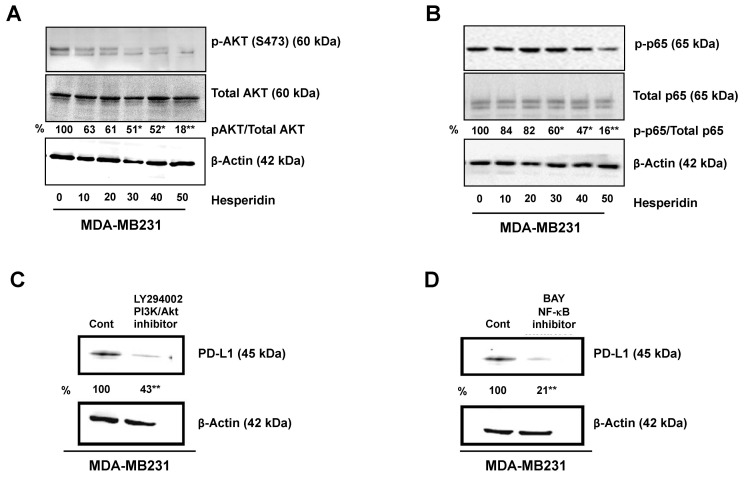
Downregulation of PD-L1 protein via inhibition of Akt and p65 phosphorylation in MDA-MB231 cells treated with hesperidin: (**A**) Phosphorylation of Akt, (**B**) phosphorylation of p65, and (**C,D**) PD-L1 protein in the absence or presence of inhibitor (PI3K/Akt LY294002 inhibitor and NF-κB inhibitor BAY) Statistical significance * *p* < 0.05 and ** *p* < 0.01.

**Figure 5 molecules-25-00252-f005:**
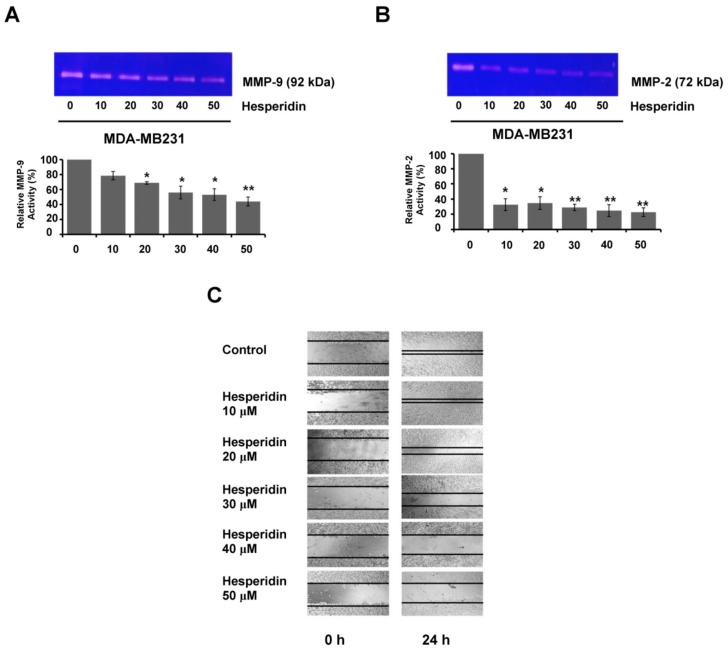
Hesperidin suppressed the migration and MMP secretion in PD-L1 high-expressing MDA-MB231 cells. The relative activities of (**A**) MMP-9 (**B**) MMP-2. Bar diagram represents the relative activity of MMPs in hesperidin treated groups compared to the control group considering the control value as 100 percent. (**C**) Hesperidin inhibited the migration in MDA-MB231 cells. Data indicated as mean ± SD from three individual experiments. Statistical significance * *p* < 0.05 and ** *p* < 0.01 versus control (non-treated cells).

**Figure 6 molecules-25-00252-f006:**
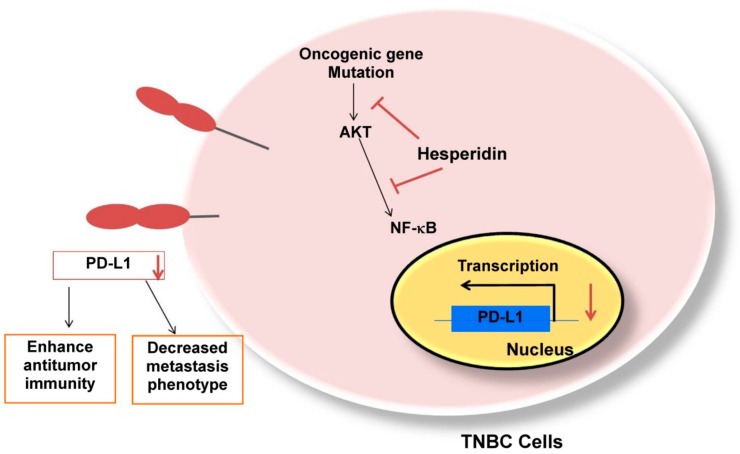
Proposed model of hesperidin suppressing Akt and NF-κB activation that mediated PD-L1 mRNA and protein expression. Second, hesperidin inhibited the migration and MMP-9 and MMP-2 secretion in MDA-MB231 cells.

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
