# Peer review of "Inhibitory Effect of Hesperidin on the Expression of Programmed Death Ligand (PD-L1) in Breast Cancer"

_molecules, 2020, doi:10.3390/molecules25020252_

Round 1
Reviewer 1 Report
I'm not able to revise the manuscript until authors produce a lot of supplementary data: the entire gels of WB tests (how they split and clearly analyse proteins at 42 and 45 KDa and fosfo and total proteins? They didn't describe the method). All raw data of replicates. Method to determine MMP is not described.
Author Response
We apologized for any inconvenience. We added the supplementary and raw data including . the figures of entire gels. of western blot analysis in the revised version o manuscript We described the methods to detect proteins at 42 and 45 KDa and fosfo and total proteins and how MMPs were determined. Please kindly see the attached raw data file.

Reviewer 2 Report
1. Fig1B - There is a lack of an error bar in MCF-7, and all the figures should be re-organized to enhance the quality of the presentation.
2. Fig2 - The layout is not satisfied and the resolution of fig2A is poor.
3. Fig3 - The error bar is missing in the control, and all other treatments have too large error bars which reflect that the variation is high between treatments.
4. Fig3B - Why PD-L1 at 0.17 only has partial band?
5. Fig4 - The quality of several bands is poor.
6. Fig5 - The layout is not satisfied.
7. A graphic summary for the whole story is needed, especially for the inhibition by hesperidin on the expression of PD-L1 via downregulation of Akt and NF-κB signaling.
8. A comprehensive comparison for the bioactivities of hesperidin in a table should be made.
Round 2
Reviewer 1 Report
Authors addressed all major concerns.Reviewer 2 Report
The MS has been improved, and it's quality is acceptable now.